# Comparison of Self-Reported and Device-Based Measured Physical Activity Using Measures of Stability, Reliability, and Validity in Adults and Children

**DOI:** 10.3390/s21082672

**Published:** 2021-04-10

**Authors:** Janis Fiedler, Tobias Eckert, Alexander Burchartz, Alexander Woll, Kathrin Wunsch

**Affiliations:** Institute of Sport and Sport Science, Karlsruhe Institute of Technology, 76131 Karlsruhe, Germany; tobias.eckert@kit.edu (T.E.); alexander.burchartz@kit.edu (A.B.); alexander.woll@kit.edu (A.W.); kathrin.wunsch@kit.edu (K.W.)

**Keywords:** self-report, device-based measured, physical activity, reliability, validity, stability

## Abstract

Quantification of physical activity (PA) depends on the type of measurement and analysis method making it difficult to compare adherence to PA guidelines. Therefore, test-retest reliability, validity, and stability for self-reported (i.e., questionnaire and diary) and device-based measured (i.e., accelerometry with 10/60 s epochs) PA was compared in 32 adults and 32 children from the SMARTFAMILY study to examine if differences in these measurement tools are systematic. PA was collected during two separate measurement weeks and the relationship for each quality criteria was analyzed using Spearman correlation. Results showed the highest PA values for questionnaires followed by 10-s and 60-s epochs measured by accelerometers. Levels of PA were lowest when measured by diary. Only accelerometry demonstrated reliable, valid, and stable results for the two measurement weeks, the questionnaire yielded mixed results and the diary showed only a few significant correlations. Overall, higher correlations for the quality criteria were found for moderate than for vigorous PA and the results differed between children and adults. Since the differences were not found to be systematic, the choice of measurement tools should be carefully considered by anyone working with PA outcomes, especially if vigorous PA is the parameter of interest.

## 1. Introduction

Insufficient physical activity (PA) is a high-risk factor for non-communicable diseases in modern society [1] and is linked to an overall increased mortality rate [2]. This in turn leads to a high economic burden worldwide [3] and calls for a systematic approach to increase PA. To counteract the insufficient PA levels, the world health organization (WHO) has continuously put PA guidelines into place [4]. One of the main challenges with these guidelines is to classify someone as sufficiently active since PA can be measured in various ways yielding different outcomes [5,6,7,8]. Unfortunately, there is no basic solution as van Hees described in a recent blog post [9]. According to him, PA is defined by the measurement method used and guidelines represent the average results of a variety of different methods which is not feasible to apply to the conception of intervention studies [9]. This is especially important if the study aims to use a personalized approach as with just-in-time adaptive interventions [10] or aims to compare the result to common PA guidelines which both strongly depend on comparable data. The previous World Health Organization (WHO) guidelines [11] were mainly based on data revealed by studies using subjective assessment methods, e.g., questionnaire data from the Global Physical Activity Questionnaire [12] and the frequently used International Physical Activity Questionnaire (IPAQ) [13] with both showing low to strong correlations (lower correlations for moderate PA (MPA) than vigorous PA (VPA)) in previous studies [14,15]. Using self-report assessments is convenient in large samples but the results are inconsistent due to either over- or underreporting of PA [5]. To counteract recall-bias, other self-report, as well as methods such as PA diaries and, with new technological advances, ecological momentary assessments are used due to their timeliness and a smaller Blackbox due to multiple measurements, which in turn increases the burden for participants due to more frequent reports [6,16,17]. The most recent WHO guidelines adapted the recommendations according to findings from studies using accelerometry, pedometer, and other device-based measurements which found that PA bouts of less than 10 min (not monitored by most PA questionnaires) also qualify to boost health benefits [4,18] while they remove the recall bias [19]. Furthermore, the rising interest in short-time intermitted VPA for health benefits with a reduced time requirement is especially difficult to monitor using self-reports [20]. However, to gain reliable measurements of PA, the sensors should be worn long enough to accurately represent the measurement duration of interest (e.g., eight hours a day for at least four days can accurately represent one measurement week) [20,21]. Even though a wear time of 24 h per day is described to be the most accurate assessment method for overall behavior throughout the day (i.e., sleep, sedentary behavior, and PA) [21], accelerometers are rarely attached to the body for this duration and therefore measured PA can be impacted by wear-time bias (a wear-time of 8 to 10 h is commonly assumed to be sufficient [21] but PA can occur during the non-wear-time and therefore PA is likely to be underestimated as compared to real PA during 24 h). While wearables such as Fitbit can easily be attached to the body for 24 h, they are mainly designed for commercial purposes, show limited validity and reliability, and can only provide accurate step counts in adults under certain conditions (no mobility limitation and worn at the torso) but not for energy estimation [22]. In accelerometers, the use of different sensors, algorithms, cut points (point which determines PA intensity), and epoch lengths (e.g., raw data, 1 s, 10 s, 60 s) used in measurement and analysis of PA has a high impact on PA estimations, depends on the age group (e.g., recommendation for the use of shorter epochs in children [23]) and complicates comparison between different studies [23,24,25]. Here, the choice of epoch length is especially important in detecting VPA and inactivity in children [23] due to their highly intermittent PA patterns [26]. Even though PA patterns in adults are often linear (i.e., less short duration and high-intensity PA) and PA, therefore is believed to be not as susceptible as children’s PA, the use of different epoch lengths alone can also change moderate to vigorous PA estimations in adults due to the smoothing of PA intensity with the use of longer epochs (i.e., 10-s vs. 60-s epochs) [23,25]. Thus, each approach has its own challenges and there is currently no gold standard to measure PA if using accelerometry, questionnaires, or diaries [27], even though best practices to handle these issues are currently discussed [21]. Comparing these measurement methods (i.e., accelerometry, diary, and questionnaire) is therefore challenging and requires further data on their relationship between each measurement method and it is important to evaluate if the relationship is consistent over time (i.e., over two measurement periods) under consideration of multiple aspects.

To gain further insights into the issue at hand, statistical quality criteria of the different methods have to be considered [28]. Thereby, test-retest reliability (in the following referred to as reliability) shows good to excellent intraclass correlation coefficients for accelerometry (for everyday activity) [29]. Furthermore, the evaluation of the reliability of self-report questionnaires (i.e., GPAQ [15] and IPAQ [13]) and a PA diary [30] also indicate good to excellent reliability, while others found poor reliability in some metrics of the GPAQ [31]. Measures of agreement (in the following referred to as validity) expressed by the correlation between self-reported and device-based measured PA, show an overall low agreement, are influenced by age and gender and self-reported results are often overreported when compared to device-based measurements especially for VPA [5,14,15]. One study which compared PA data measured by accelerometry, diary, questionnaire, and interview in adults (*n* = 1916) found that the comparison between the device-based measured and self-reported meeting of PA recommendation at one measurement period yielded only 12% agreement based on pairwise comparisons [6]. Other studies that analyzed test-retest reliability at several timepoints and included diaries [32,33] or accelerometer [34,35,36] to analyze the validity at one or all measurement periods also reported good test-retest reliability but only acceptable or comparable validity showing that comparing PA results of different measurement methods should be done with caution. These discrepancies indicate the difficulty of the interpretation for sufficient PA using different methods.

However, even though differences between measurement tools are frequently reported throughout several studies [6,7,8,27], there are, to the best of our knowledge, currently, no studies analyzing whether these differences are systematic (i.e., high correlation for the paired differences between the measurement methods between two separate measurement weeks) for PA levels measured via accelerometry, diary, and questionnaire for adults, children, and adolescents (in the following referred to as children). If these differences could be shown to be consistent over time, this would strengthen the interpretation and comparison, and use of different PA data in intervention studies, in between different studies, and regarding PA guidelines. Here, longitudinal data may represent a more consistent picture of PA, allowing to detect time-stable differences regarding the amount of PA between the different methods.

Therefore, the current study aimed to examine the stability of the pairwise differences between three PA measurement methods (i.e., accelerometry, diary, and questionnaire) and the influence of different evaluation techniques (i.e., epoch lengths of 10 s and 60 s for accelerometer data for MPA and VPA in adults and children between two independent measurement weeks in an explorative manner. A secondary aim was to analyze the reliability of the above-mentioned measurement methods and to assess the validity of those methods.

## 2. Materials and Methods

### 2.1. Participants and Procedure

Participants were eligible for this study if they represented a family with at least one child and one adult who were living in a common household. In total, 74 adults and 74 children participated in the SMARTFAMILY (SF) trial which consists of a theory- and evidence-based mHealth intervention and targets health behavior change in families (further information are described in the study protocol [37]) and all participants of the control group (32 adults age 37–55 years and 32 children age 5–19 years) were eligible for the present study. Full ethical approval was obtained for SF. All participants, children, and legal guardians provided written informed consent prior to commencing the study by signing the informed consent form (The International Registered Report Identifier (IRRID) for the SF study is RR1-10.2196/20534.). The trial was conducted in accordance with the Declaration of Helsinki.

Participants were recruited in schools, school holiday programs, music schools, sports clubs, via personal communication and via newspapers and email distribution lists. Participants were cluster-randomized to an intervention group and a control group. Whereas the intervention group received a three-week mobile health intervention between the two measurements, families of the control group had a three-week waiting period without any intervention. Baseline (T_0_) and post-intervention (T_1_) data of the control group were used for this study because the intervention might have influenced PA sampling at T_1_. Data collection at T_0_ and T_1_ consisted of the measurement of PA by accelerometer, diary, and questionnaire over one week which was identical for T_0_ and T_1_ (which were at least three weeks apart). For children, the inclusion of questionnaire data was not feasible for this study due to the use of a questionnaire without the indication of minutes per week for PA (Sixty-Minute Screening Measure [38]) which is also not comparable to the new PA guidelines which recommend an average of 60 min PA per day for children [4].

### 2.2. Measurements

#### 2.2.1. Accelerometer

Hip-worn (right side) 3-axial accelerometers (Move 3/Move 4, Movisens GmbH, Karlsruhe, Germany) were used to continuously record PA (see Appendix A). These accelerometers are scientific research instruments with a measurement range of ±16 g, an output rate of 64 Hz, physical dimensions of 62.3 mm × 38.6 mm × 11.5 mm, weight of 25 g, and custom epoch lengths (i.e., 10 s and 60 s). Data is recorded in a rare format (64 Hz) and afterward summarized in the epoch lengths of choice. Epoch lengths were chosen to represent the most common used epoch length (60 s) which was mainly used due to limited storage in the past, and a shorter epoch length (10 s) as shorter epoch lengths are believed to be more appropriate to estimate VPA and to assess PA in children due to intermittent movement behavior [21]. Validity has been evaluated for a previous version of the accelerometer (Move 2) which uses comparable digital signal processing as the move 3/4 [39] and has been considered accurate for assessing steps [40] and energy estimation [19,41] in adults. Handling of the accelerometer was explained and demonstrated by a study instructor and participants were instructed to wear the accelerometer during wake-time and to remove it only for taking a shower, swimming or during certain sports involving bodily contact to minimize the probability of injuries. Outcomes for the accelerometer which were used for this study were MPA (3.0–5.9 MET) and VPA (≥6 MET) (light PA was not considered because the questionnaire has no comparable measure) for all participants. MET values were calculated based on activity class (based on acceleration and barometric signals) which determines the estimation model. Afterwards, movement acceleration, altitude change, and demographics were combined in the model for the MET estimation [41] (see Appendix A).

Accelerometer data were included if a minimum wear time of at least 8 h a day for at least 4 of the 7 days during the measured week was obtained. Non-wear time was calculated on the accelerometer in 30-s intervals. The non-wear time detection was based on an algorithm that used accelerometry and temperature signal over a 10-min window to distinguish between wear time, non-wear time, and sleep as described elsewhere [42]. For valid measurements, the average of MPA and VPA per valid day was multiplied by 7 to represent the total minutes per week.

#### 2.2.2. Diary

A daily PA diary was filled in by all participants complementary to wearing accelerometers indicating date, time and type of activity, duration, and perceived intensity on every single day within the two measurement weeks. Each activity was recorded separately and participants were instructed to rate the respective PA intensity as either light (no perspiration or shortness of breath), moderate (some perspiration and shortness of breath) or vigorous (profound perspiration and shortness of breath). Participants were asked to report all PA with a duration of more than 10 min. Analogous to accelerometry outcomes, MPA and VPA were summarized as total minutes per week.

#### 2.2.3. Questionnaire

At the end of each measurement week, adults were asked to fill in the German short version of the IPAQ [43] which is available at the IPAQ website [44], asking retrospectively for activities during the previous week. The results of the question relating to minutes spent in MPA (comprising of moderate activity and walking [45]) and VPA were calculated for this study by multiplying the reported amount of days with the reported duration of the indicated activity per day. Therefore, the outcomes MPA and VPA were also recorded as total minutes per week. Children completed the Sixty-Minute Screening Measure [38] for moderate to vigorous physical activity which yields binary results (sufficiently active vs. insufficiently active according to the previous WHO guidelines [11]) and was not included in this study to maintain total minutes per week as a unit. Therefore, all results referring to the questionnaire are limited to adults. Additionally, questions about age and anthropometry were included in the questionnaire among others (see the study protocol for detailed information [37]).

### 2.3. Statistical Analysis

To compare the mean differences for the four PA measures (i.e., accelerometry with 10 s and 60-s epoch lengths, diary, and questionnaire) between T_0_ and T_1,_ the differences between both measurement time points were calculated in total minutes per week for MPA and VPA for all combinations (i.e., six combinations for adults and three combinations for children) and defined as new parameters (ranging from −607.17 to 398.29 min/week) at each measurement week. If one of the original parameters included missing data, the parameter expressing the difference was also considered as missing data for the participant. Additionally, test-retest reliability for each parameter between T_0_ and T_1_ and a validity measure by pairwise comparison of all parameters at both T_0_ and T_1_ were calculated (see Figure 1).

The raincloud plots have been created using RStudio [46] and the ggplot2 package [47], following the instructions of Allan and colleagues [48]. Statistical analyses were performed in RStudio using the RVAideMemoire package [49]. Descriptive characteristics of all participants are displayed as means with standard deviation (SD). The degree of agreement for all calculations was assessed using the Spearman correlation coefficient (*r_s_*) by the cor.test() function in RStudio since the data differed significantly from a normal distribution in the Kolmogorov Smirnov test, which was confirmed via visual inspection of the distribution in histograms. First, *r_s_* values between T_0_ and T_1_ were calculated between the pairwise differences of all parameters to indicate the stability of these differences (main aim). Then, *r_s_* values between T_0_ and T_1_ were computed for each parameter separately to indicate test-retest reliability (secondary aim). Afterwards, *r_s_* values for the pairwise comparison between all combinations of parameters at both T_0_ and T_1_ were computed for a measure of validity (secondary aim). Afterward, Confidence intervals were added by using bootstrapping (*n* = 1000). All calculations were performed for children and adults separately and pairwise deletion was used for each calculation.

*r_s_* were interpreted under consideration of the 95% confidence intervals as recommended by Schober, Boer, and Schwarte [50]. The level for significance was set a priori to 0.05 and was based on the correlation and not on the confidence intervals.

## 3. Results

### 3.1. Participant Characteristics

The data of 32 adults and 32 children was used in this study. Characteristics of the participants are presented in Table 1.

### 3.2. Physical Activity Outcomes

The descriptive results of PA measurements at T_0_ and T_1_ and corresponding reliability and validity measures (*r_s_*) are presented in Appendix A. Figure 2A,B visualize the descriptive PA level measured by each measurement tool for adults and Figure 2B,D for children. Overall, the descriptive values show the highest PA values for the IPAQ, followed by accelerometry with 10-s epochs and 60-s epochs, and the lowest PA values are reported for the PA diary. These results are consistent for MPA and VPA in both adults and children except for VPA in children where the PA diary shows the highest PA values. MPA in T_0_ is higher in all measures compared to T_1_ whereas VPA values are only consistently lower in children at T_1_. 

#### 3.2.1. Stability between the Differences of the Parameters at the Two Measurement Weeks

Table 2 presents the *r_s_* for the differences in minutes per week of all parameters compared from T_0_ to T_1_ for adults while Table 3 shows the results for children.

The differences in the amount of PA gathered by accelerometers using 10 s and 60-s epoch lengths showed a significant relationship for both adults and children in MPA and VPA between T_0_ to T_1_.

Significant associations of the differences between accelerometry and diary were found for MPA, but not for VPA, measured by 10-s epochs, and PA diary for adults. For children, there was a significant relationship between the differences of accelerometry using 10-s epochs and the PA diary for VPA, but not for MPA, between T_0_ and T_1_ with a lower confidence limit below zero.

The differences between accelerometry and the IPAQ were significantly related for both 60 s and 10-s epochs concerning MPA but not for VPA.

No significant association at all was found for the differences of the diary and IPAQ between T_0_ and T_1_.

#### 3.2.2. Test-Retest Reliability

Adults MPA measured by accelerometry at T_0_ shows a significant relationship with MPA measured by accelerometry at T_1_ with the lower confidence interval limit of *r_s_* > 0.5 at both epoch lengths (see Appendix A). For children, only VPA at 10-s epochs showed a similar effect. The other accelerometry measures also show significant correlations but lower confidence limits.

For the diary-based PA, adults PA has no significant relationship between T_0_ and T_1_ while childrens’ PA shows a significant relationship with the lower confidence limit of around 0.

PA measured by the IPAQ at T_0_ has a significant relationship with PA measured by the IPAQ at T_1_ for MPA with a lower confidence limit of around 0.1.

#### 3.2.3. Validity

Additional analysis of pairwise *r_s_* between all methods at each T_0_ and T_1_ showed a significant relation between 10 and 60-s epochs at T_0_ and T_1_ for both children and adults with the lower confidence limit above 0.7 for both MPA and VPA (see Appendix A). The IPAQ showed a significant relationship to accelerometry for VPA (compared to 10-s epochs and PA diary) with lower confidence limits of around 0 only at T_0_. No further significant relations were found between the parameters at neither measurement week.

## 4. Discussion

This study aimed to examine the reliability, validity, and stability of a PA questionnaire, a PA diary, and accelerometry using 10 and 60-s epochs for MPA and VPA in adults and children over two measurement weeks. The main result evoked the stability of differences to be an interesting additional measure for the comparison of different measurement methods not necessarily being in concordance with reliability and measures of validity. Overall, descriptive results consistently showed that self-reports via questionnaire revealed by far the highest PA amounts, followed by accelerometry with 10 and 60-s intervals. The lowest amounts were detected for PA measured via diary for both MPA and VPA in adults and children with a large variance in the results of each measurement tool. Only device-based measured PA showed reliable, valid, and stable results for the two measurement weeks for both epoch lengths. The IPAQ yielded mixed results and the PA diary showed few significant relations for stability in adults and mixed results in children.

### 4.1. Quality Criteria

#### 4.1.1. Stability

The comparison of the pairwise differences between T_0_ and T_1_ showed stable results for almost all comparisons in adults’ MPA. For children, stable MPA differences were found for 10 and 60-s epoch lengths and the diary (which did not reach statistical significance but indicates *r_s_* of 0.4). VPA mainly showed a significant correlation between the two epoch lengths in adults and children, while 10-s epochs were only associated with the diary in children. Taken together, the stability results showed significant results for all parameters also demonstrating high reliability and validity, which was to be expected. No stability was found for parameters with low reliability and significant validity (i.e., adults VPA 10 s epochs to IPAQ and diary to IPAQ). However, some measures also showed significant stability where no validity, and in one case where neither reliability nor validity, was found to be significant. This indicates the importance of the relationship over time because these results would have been missed without a stability measure.

These results show that the relation between the measures including self-report differed between the measurement weeks and are therefore not stable over time which gives reason for concern in the comparability of these measures (as indicated by validity as well). However, the comparison of device-measured PA and the diary in children indicates some stability which might show that the diary is more feasible for children than for adults and strengthens the point that children’s structured PA might be easier to determine using self-report. Additionally, the descriptive values showed that only the comparison of 10 to 60-s intervals in children yielded minimum and maximum values without a change of signs indicating that these differences were exposed to intraindividual variations for most of the device based measured results and are not 100% consistent even though they are highly related (i.e., comparison of 10 to 60-s intervals). These findings have to be treated with caution and need to be reevaluated due to the limitations listed below.

#### 4.1.2. Reliability

Test-retest reliability of both epoch lengths indicated that the present data of two measurement weeks represented comparable weeks of everyday life concerning PA. This was partially confirmed by the IPAQ (only for MPA) and the PA diary (only for children). This finding differed from other studies which found the IPAQ and PA diaries to also yield reliable results [13,30]. The true amount of PA remains unknown, as is the case with all estimates, but the reliability of accelerometry can be seen as a benchmark indicating that both weeks are comparable. This, however, thrives the question of why the self-reported measures showed limited reliability in our sample. One reason could be that the perception of PA load changed for participants between T_0_ and T_1_, e.g., because they were bored of the repeated questions or they reflected more about their PA the second time. The reason why children’s PA diaries show some reliability might be that they showed higher VPA in all comparable measures. This might be indicative for the circumstance that the children in our sample engaged to a high amount in structurally organized VPA (e.g., training in a sports club, school sport lessons) which is easier to document in a diary than short and intermittent bouts of occasional VPA during everyday situations (e.g., playground), which was previously reported to represent the nature of VPA in children [23]. Finally, the actual PA might also have changed, even though the device-measured results were reliable, which can, however, not be evaluated in the present study because no gold standard of PA measurements has been assessed.

#### 4.1.3. Validity

Unsurprisingly, PA evaluated by 10 and 60-s epoch showed a high and consistent validity among each other for both MPA and VPA at T_0_ and T_1_ for adults and children despite their indicated total amount of PA differed descriptively. Total values were consistently higher for 10-s intervals than for 60-s intervals, depended on the population and PA intensity (differences: adult MPA: 42–47%, VPA: 46–82%; children: MPA: 14–18%, VPA: 50–56%), even though adults are thought to have longer intervals of PA which should be stable for different epochs [25]. To illustrate this issue with an example: If a person is moving up one level of stairs rather fast in 20 s and stops at the top to have a conversation with a colleague, the use of 10-s epoch length would detect 10 to 20 s of VPA while the use of 60-s epochs would calculate the mean over this longer time period and end up with light or MPA (the total metabolic equivalent (MET) would not differ between the epochs, but classification would). Therefore, the high changes in MPA for adults, in this case, may have arisen from a switch of MPA to light PA in the longer epochs because of multiple occasions where MPA lasted less than one minute (e.g., walking short distances in the office).

This supports the importance to consider the impact of epoch lengths on PA outcomes as mentioned in previous studies [23,25]. Here, in line with the credo of “every move counts” [4] it is recommended to choose shorter epoch lengths as these might capture short bouts of MPA and VPA more sensitively than longer epoch lengths. Concerning the validity of the self-reported measures, the IPAQ was associated with 10-s epochs, and the PA diary at T_0_ for adults’ VPA and no further comparison of measurement tools showed a significant result, not supporting weak correlations found in previous studies [8,13]. Descriptive results from the PA diary indicate the lowest reported MPA and VPA (except for children VPA), while the IPAQ showed the highest PA results, which was also found by Hukkanen and colleagues in adults [6]. One reason for the difference between the self-report measures in the current study could be that participants were instructed to classify their PA in the diary as light, moderate or vigorous while only MPA and VPA were included in this study. Since the IPAQ, has no measure for light PA, participants might have classified their light PA in the diary as MPA in the IPAQ. This might be responsible for the high MPA values reported by the IPAQ but this does not explain the high discrepancy in results for VPA in adults where the PA diary also showed the lowest values. This implies the importance that all measurement methods in a study include the same outcome variables, especially if the measurement methods are compared to each other. A further complication with the PA diary was that there has been no indication included if the diaries have been filled out daily or at the end of the week and that there was no distinction between missing values and no PA. Furthermore, indicating if the accelerometer was worn during the PA which was documented in the diary would have allowed a more detailed impression of discrepancies and the true value of PA during the week. This will be accounted for in the SF2.0 study [37].

### 4.2. General Discussion

The results of the current study are mainly in accordance with the current literature indicating higher reliability than validity for the three measurement tools [5,6,7,8,14,15,27,30] and revealing that the epoch length influences PA estimations [23,25]. In contrast to earlier results, we found limited reliability for the PA diary and the questionnaire in adults’ VPA. The inclusion of stability shows more stable results in children than in adults, especially for the diary, and adults’ VPA is only stable if the two epoch lengths are compared. This has important implications for the use of these measurement tools. Based on current results, future research should further explore the stability between different measurement tools over time to gain further knowledge about the relationship, trying to find a solution to compare single measurement methods to the mixed-method approach in the WHO guidelines. Moreover, different assessment methods should be used which can complement each other such as ecological momentary assessment and accelerometry. Researchers should be aware of the limitations of and within each measurement tool and ensure that it is the best fit for the purpose in question. Hence, differences between adults and children in PA research should be considered to deepen the understanding of these differences. Future studies should also aim to create comparable data sets with clear and thorough reporting of outcomes to enable the merging of data in order to be able to compare more subgroups and different settings. Here, it might be helpful to provide a relative amount of PA compared to the wear time or 24-h measurements in order to compare results between participants in greater detail in future studies. In order to confirm and refine our findings, a replication study with data from the SF2.0 trial with a feasible questionnaire for children and the GPAQ for adults will be conducted in the future [37].

### 4.3. Strengths and Limitations

The main strength of the current study is the concomitant use of all measurement methods (i.e., accelerometry, questionnaire, and diary) within the same time frame and that they were repeated in the same manner without any intervention in between. Furthermore, both measurement weeks represented an everyday week (i.e., no measurements during holidays), which enhanced comparability. Including reliability and validity as secondary aims in this study helped to interpret and understand the stability results more accurately. This is especially important as results showed reliability and validity to differ from stability results in some cases. Furthermore, the inclusion of data from both adults and children allowed us to analyze differences between these populations. Here, further distinctions between children and adolescents will be interesting to examine in a larger sample. Finally, the use and detailed reporting of multiple measurement tools strengthened the explanatory power of results and allowed for comparison with existing research.

However, there are some limitations to mention. First of all the true value of PA is unknown and each measure is just an estimate of PA as, for example, no 24-h measurements of energy expenditure via indirect calorimetry or throughout observation of activity patterns have been recorded [21]. Evaluating the relationship between the different parameters is even more complicated as most questionnaires and PA diaries only ask to report PA with a duration of at least 10 min (or even asking for PA over or under 60 min [38]). With the new WHO guidelines for PA [4], self-reports have to be adapted to indicate guideline adherence in larger samples and to be comparable to accelerometry data [18]. This might be achieved by removing the wording of reporting only 10 min of PA which, however, would increase participant burden and limit adherence due to more detailed reporting requirements and the possible benefit will have to be evaluated in future trials [18].

Another limitation is the rather small sample size which is further divided into adults and children. This limits the generalizability of results and the exploration of subgroups e.g., divided by gender or evaluating results for children or groups of different PA levels separately. Comparisons were also limited as there was no feasible questionnaire included for children in this study. Furthermore, the four-day measurement criteria including eight hours a day might have impacted the measured PA values even though it is assumed to be a sufficient measurement duration [51], increased the convenience for participants, and allows for reduced loss of data while maintaining reliable data [52,53]. Furthermore, sedentary behavior was not included in this study, even though the updated WHO recommendations include these important measures [4]. However, hip-worn accelerometers only capture inactive behavior, but not sedentary patterns (e.g., sitting, lying [54]) as has been discussed elsewhere [55]. To gain a fair impression of these parameters and to cover all 24 h of the day, future studies should include the comparison of the outcomes for sedentary behavior and light PA under consideration of non-wear time within a 24-h measurement approach to evaluate shifts between physical activity levels (e.g., if a higher amount of VPA occurs due to less non-wear time or less SB) [56].

Finally, because data differed significantly from a normal distribution and especially VPA was skewed due to many low values, no intraclass correlation coefficients could have been calculated, which would have been more accurate as they comprise the total mean value of the measure in the equation [57,58]. Due to the large number of comparisons, the use of Bland Altman plots as an alternative method for such comparisons [59] would be fairly interpretable and was therefore not feasible in our study which used an explorative approach. Future studies should consider a more specific approach with fewer comparisons by formulating clear hypotheses for the present results (e.g., stability for MPA in adults) and use Bland Altman plots to analyze the data in greater detail.

## 5. Conclusions

Based on the results of the current study, a comparison between PA estimations (especially for VPA) measured by different tools should be carried out with caution and only if all measurement methods include the same outcome parameters over the same period of time. Here, it needs to be stressed that everyone working with PA values (e.g., scientists planning and conducting PA studies, practitioners giving detailed health-related PA advice, and consumers trying to estimate if they are sufficiently active compared to the guidelines) should carefully consider the measurement tool to be suitable for the purpose in question because considerable discrepancies in results can be detected. Furthermore, it is crucial to use standardized reporting to enhance the comparability of the data (e.g., for future meta-analyses) [60]. Finally, self-reported measures can offer additional contextual information of PA in a timely manner by using e.g., ecological momentary assessments [17,61] to further refine our understanding of PA and may lay the foundation for personalized intervention approaches as with just-in-time adaptive interventions [10] in the future.

## Figures and Tables

**Figure 1 sensors-21-02672-f001:**
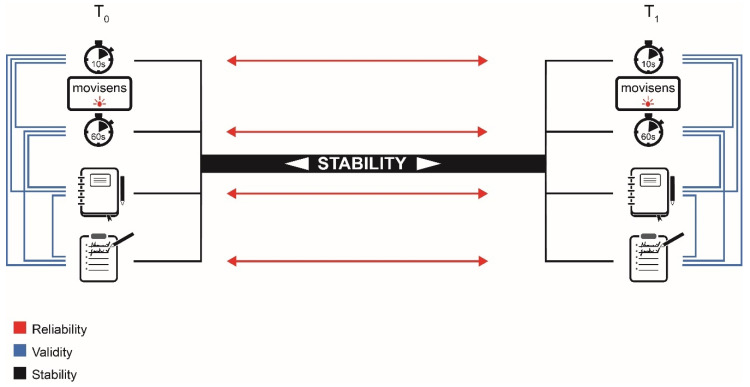
Study design. Displayed are the calculated combinations for validity (blue brackets) and reliability measures (red arrows) for the secondary aims concerning the parameters (from top to bottom) accelerometry using 10-s epochs and 60-s epochs, a physical activity diary, and the International Physical Activity Questionnaire. The main aim consisted in comparing the difference in total minutes for each bracket from T_0_ to T_1_ (black).

**Figure 2 sensors-21-02672-f002:**
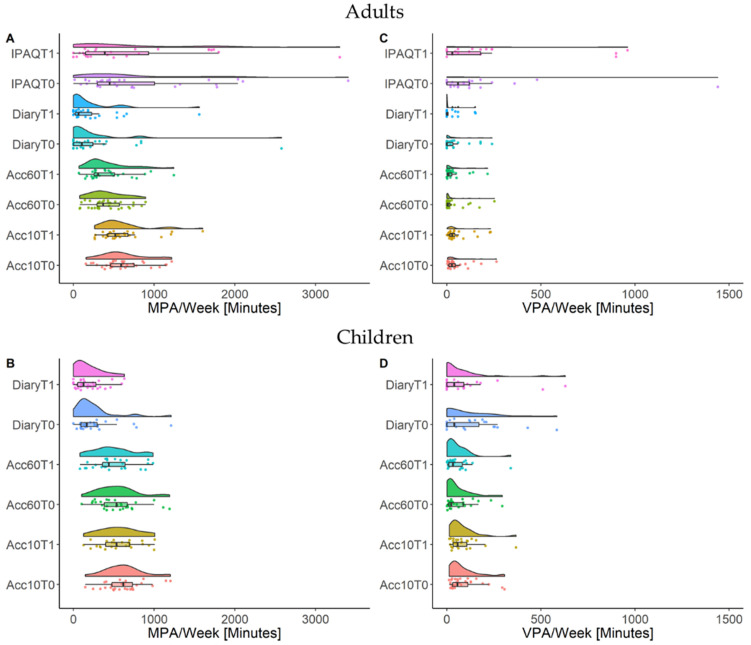
Descriptive means of moderate physical activity (MPA) in adults (**A**) and children (**B**) as well as vigorous physical activity (VPA) in adults (**C**) and children (**D**). Displayed are the results (independent measurements, distribution, and box-plots) of the physical activity diary (Diary), accelerometry with 60-s epochs (Acc60), accelerometry with 10-s epochs (Acc10), and the International Physical Activity Questionnaire (IPAQ) for two independent measurement weeks (T0 and T1) in minutes per week.

**Table 1 sensors-21-02672-t001:** Characteristics of the participants. Displayed are the number of participants (N), means, and standard deviations (SD) for the parameters gender (male/female), age in years (y), height in centimeter (cm), and weight in kilogram (kg).

Parameter	Adults		Children	
	N	Mean (SD)	N	Mean (SD)
Gender (m/f)	11/21	-	15/17	-
Age (y)	31	47.90 (4.44)	32	13.22 (2.94)
Height (cm)	31	170.42 (8,52)	31	162.68 (17.61)
Weight (kg)	31	72.74 (13.27)	29	51.10 (14.10)
BMI (kg/m^2^)	31	24.97 (3.62)	29	18.96 (2.94)

**Table 2 sensors-21-02672-t002:** Results of stability of differences calculated by Spearman’s rho (*r_s_*) for adults. Displayed are the number of participants (*n*), mean differences between the measurement tools (Mean), their standard deviations (SD), and their minimum (Min) and maximum (Max) values at two measurements three weeks apart (T_0_ and T_1_) as well as the differences in the error between T_1_ and T_0_ as mean, SD and percentage (%). Additionally, *r_s_* with corresponding *p*-value (* for *p* < 0.05) and 95% confidence interval via bootstrapping (CI) for differences between the measurement tools: accelerometry with 60 s epoch length (Acc 60) and 10 s epoch length (Acc 10), physical activity diary (Diary), the International Physical Activity Questionnaire (IPAQ) for moderate (MPA) and vigorous (VPA) physical activity.

	Adults				
	*n*	T_0_ Mean (SD) [Min-Max]	T_1_ Mean (SD) [Min-Max]	T_1_-T_0_ Mean (SD) Difference [%]	*r_s_* (*p*-Value) [CI]
	**MPA (min/wk)**				
Acc 10–Acc 60	28	206.47 (74.33) [122.50–456.40]	181.40 (102.78) [−207.67–364.00]	−25.06 (107.39) [−12.93%]	0.604 (0.001 *) [0.242–0.842]
Acc 10–Diary	25	394.80 (533.57) [−1513.00–1152.20]	398.29 (514.08) [−1196.60–1566.83]	3.49 (70.37) [1.41%]	0.579 (0.002 *) [0.111–0.879]
Acc 10- IPAQ	25	−195.42 (764.58) [−2262.00–873.00]	−125.04 (730.91) [−2744.00–631.50]	70.37 (854.72) [43.92%]	0.613 (0.001 *) [0.268–0.841]
Acc 60–Diary	25	185.97 (528.88) [−1839.00–755.00]	216.77 (488.43) [−1324.80–1208.67]	30.80 (595.35) [15.30%]	0.699 (<0.001 *) [0.314–0.890]
Acc 60–IPAQ	25	−393.48 (779.17) [−2558.00–654.00]	−303.05 (769.65) [−2978.00–622.83]	−107.63 (852.18) [25.97%]	0.598 (0.001 *) [0.266–0.828]
Diary–IPAQ	24	−607.17 (858.53) [−3224.00–480.00]	−466.08 (743.78) [−2640.00–750.00]	−141.08 (1061.75) [26.29%]	0.072 (0.737) [−0.431–0.603]
	**VPA (min/wk)**				
Acc 10–Acc 60	28	15.39 (11.79) [−16.00–43.00]	23.37 (31.34) [1.00–176.17]	7.98 (32.51) [41.18%]	0.569 (0.002 *) [0.162–0.858]
Acc 10–Diary	25	22.74 (87.04) [−222.00–204.00]	35.50 (67.60) [−119.67–228.67]	12.75 (99.56) [43.82%]	0.197 (0.344) [−0.249–0.558]
Acc 10–IPAQ	26	−79.57 (292.69) [−1417.00–129.00]	−117.10 (281.61) [−942.50–228.67]	−37.53 (266.22) [38.17%]	0.279 (0.167) [−0.113–0.625]
Acc 60–Diary	25	6.93 (86.72) [−239.00–193.00]	10.86 (52.11) [−130.17–157.00]	3.93 (89.12) [44.18%]	0.155 (0.459) [−0.266–0.558]
Acc 60–IPAQ	26	−94.96 (292.49) [−1432.00–118.00]	−140.26 (275.82) [−957.67–52.50]	−60.67 (262.90) [38.52%]	0.142 (0.490) [−0.233–0.509]
Diary–IPAQ	25	−99.44 (289.09) [−1440.00–120.00]	−152.92 (292.10) [−960.00–17.00]	−53.48 (261.13) [42.38%]	0.219 (0.293) [−0.221–0.625]

**Table 3 sensors-21-02672-t003:** Stability of differences as calculated by Spearman’s rho (*r_s_*) for children. Displayed are the number of participants (*n*), mean differences between the measurement (Mean), their standard deviations (SD), and their minimum (Min) and maximum (Max) values at two measurements three weeks apart (T_0_ and T_1_) as well as the differences in the error between T_1_ and T_0_ as mean, SD, and percentage (%). Additionally, *r_s_* with corresponding *p*-value (* for *p* < 0.05) and 95% confidence interval via bootstrapping (CI) for differences between the measurement tools: accelerometry with 60-s epoch length (Acc 60) and 10-s epoch length (Acc 10), physical activity diary (Diary) for moderate (MPA) and vigorous (VPA) physical activity.

	Children				
	*n*	T_0_ Mean (SD) [Min–Max]	T_1_ Mean (SD) [Min–Max]	T_1_-T_0_ Mean (SD) difference [%]	*r_s_* (*p*-Value) [CI]
	**MPA (min/wk)**				
Acc 10–Acc 60	24	78.77 (48.73) [−16.33–171.50]	61.10 (54.36) [−69.00–177.00]	−17.67 (37.36) [25.27%]	0.785 (<0.001 *) [0.548–0.920]
Acc 10–Diary	23	376.80 (348.08) [−470.00–1037.83]	342.78 (326.38) [−272–979.00]	−34.02 (402.33) [9.46%]	0.363 (0.090) [−0.152–0.806]
Acc 60–Diary	23	299.96 (364.70) [−538.00–1027.33]	283.99 (331.10) [−300.75–967.00]	−15.97 (380.57) [5.47%]	0.383 (0.072) [−0.083–0.754]
	**VPA (min/wk)**				
Acc 10–Acc 60	24	29.43 (19.36) [9.33–102.67]	25.57 (21.25) [5.60–105.00]	−3.86 (12.81) [14.04%]	0.448 (0.028 *) [0.010–0.762]
Acc 10–Diary	23	−37.24 (174.17) [−541.25–295.17]	−16.02 (157.83) [−550.67–131.83]	21.23 (207.55) [79.68%]	0.417 (0.049 *) [−0.053–0.786]
Acc 60–Diary	23	−67.17 (167.10) [−564.00–234.50]	−40.83 (152.54) [−560.00–105.00]	26.35 (203.06) [48.78%]	0.275 (0.205) [−0.227–0.659]

## Data Availability

Data is available upon request.

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
