# Peer review of "Comparison of Self-Reported and Device-Based Measured Physical Activity Using Measures of Stability, Reliability, and Validity in Adults and Children"

_sensors, 2021, doi:10.3390/s21082672_

Round 1

Reviewer 1 Report

SUMMARY

The present study investigated several questions: How the differences between two methods for measuring physical activity may vary over time – over 3 to 4 weeks (“stability”)? What is the test-retest reliability of the investigated methods using a 3/4-week period between the measurements? What is the relationship (“validity” according to the authors) between the investigated methods at different time points (three/four weeks apart)?

The main contribution of this paper is to provide new information regarding how the difference between physical activity measurement methods (in particular, questionnaires vs accelerometry) may vary over time. This paper may be viewed as an illustration of the effect of the method reliability on the change in discrepancy between two methods over time.

GENERAL COMMENTS

I have found the present study quite interesting but I think there is sometimes a lack of appropriate focus on the information to be delivered to the reader and that the presentation of the analysis, results, and discussion sections could be presented in a better order in view of the primary aims of the study. In the introduction, there are a lot of (interesting) information but the authors could  more directly put the focus on the main scientific problem that justify the present study. In both statistical analysis, results, and discussion sections, the order of what is presented (reliability, validity, stability) does not appear logical to me because the focus should be on stability first, and thus this should be evocated in first position. Overall, I have also a lot of comments that reflect a lack of precision in the presented literature and some methodological concerns. Finally, I think the authors should indicate in their objectives the exploratory nature of their work and not wait for the end of the manuscript to recognize this important point.

MAJOR COMMENTS

I think the sentence “Even though a wear time of 24 hours per day is described to be the most accurate assessment method for PA [21]” (lines 57-58)  is an overstatement in comparison with what the cited authors actually wrote. A 24-h measurement has been described as an ideal strategy to assess the whole behavior (physical activity, sedentary behavior, and sleep) and to ensure better compliance (and then to likely have a more accurate picture of the physical activity behavior). However, measurements of physical activity during less than 24h per day (eg, during awake time) are not less accurate than 24-h measurements per se because a rigorous and correct use of the device during awake time could capture the physical activity behavior of the day, this without the need of using  graphical analysis nor an algorithm to detect,  with more or less error, the data to be analyzed during the 24-h cycle (i.e., data related to awake time, not nighttime sleep). Thus, I think this statement should be rewritten or removed because it provides an inaccurate view of what is known.

I found that the sentence “can only provide accurate step counts if compared to validated accelerometers [22]” (lines 63-64) is quite weird because the accuracy of a device is a feature that can be only determined using a gold-standard method. All other possible comparisons are related to the agreement between devices/methods. Please consider another presentation of this idea.

The idea proposed in the sentence “there is currently no gold standard to measure PA [27]” (lines 74-75) is partially wrong. First, the cited reference does not support this idea but the idea that “Neither the questionnaire nor the PAM accelerometer is a gold standard for measuring PA” (p.7), which is true, but this is different from claiming that there is no gold standard at all. Second, physical activity is a complex and multi-dimensional construct (Bassett et al., DOI: 10.1249/MSS.0000000000000468), and its assessment implies to make some choices regarding the investigated components (eg, measuring the domains, and/or the dimensions (eg, intensity)) and regarding the metrics to use, and there are several gold standard methods for measuring these metrics: eg, direct observation for the activity types or steps, indirect calorimetry for physical activity intensity, etc (cf Kozey Keadle, DOI: 10.1249/JES.0000000000000206). Thus, this sentence should be rewritten to comply with what is currently accepted as a gold standard for assessing physical activity behavior.

There are several problems with the following sentences: “Thereby, test-retest reliability (in the following referred to as reliability) shows good to excellent intraclass correlation coefficients for accelerometry [29–31]. Furthermore, the evaluation of the reliability of self-report questionnaires (i.e. GPAQ [15] and IPAQ [13]) and a PA diary [32] also indicate good to excellent reliability.” (lines 80-84).

Regarding the accelerometry studies, they have not all implemented a test-retest procedure. The study by Aadland (2015) investigated INTER-device reliability and contralateral hip difference (which is different from test-retest). Barreira et al. (2015) sought to determine a minimum number of days to reliably describe physical behavior in children. Sirard et al. (2011) conducted a test-retest study for the measurement of usual physical activity over a week. This last kind of study assesses the reliability of both the accelerometer and the measured behavior (thus, not the reliability of the devices only). Thus, reliability studies may be very different depending on what kind of reliability is actually assessed. Additional details should be added when presenting the studies (I think the study by Sirard et al. is the most appropriate here among the three cited studies; but there are a lot of other studies of this kind).

Regarding the studies cited for the questionnaires, the authors should be more nuanced in their statement for several reasons: first, the cited studies provide a partial (and thus biased) view of the literature (eg, the study by Riviere et al.; DOI: 10.1016/j.jshs.2016.08.004, found a poor reliability for some metrics with the GPAQ). Second, the reliability of a questionnaire is very likely to depend on the investigated metrics, and a general statement (e.g., “questionnaires are good”) is likely to be inappropriate because questionnaires perform differently according to the behavior intensity that is assessed.

There are some concerns regarding the references cited for the validity of the move 3 / 4 accelerometer. The authors cited Anastasopoulou (2014) and have claimed, based on the cited article, that the device they used was “accurate for assessing PA” (lines 141-142). A first problem is that Anastasopoulou et al. used a move II device. Can the present authors be sure that the results for the move II device could be the same as for a move 3 /4 device? Moreover, Anastasopoulou et al. concluded that the move II is more appropriate than the GT3X for estimating energy expenditure, but they did not claim that the move II device was accurate per se (and actually, what we can only say is that a device is sufficiently accurate for a given purpose based for example on clinical criteria, rather than claiming that a device is absolutely accurate. The present authors also cited the manufacturer (movisens)’s website when claiming that a device “is accurate” but it would be more appropriate to cite scientific first-hand sources concerning the appropriate device. Please consider these comments to formulate a correct statement in relation to the actual literature.

Lines 140: It was not clear why the device recorded the data using two epoch lengths (10 s and 60 s). Can the authors explain this point?

Lines 153: Please justify/comment the choice of using a window of 20 min to determine non-wear time. Previous work (Choi et al., 2011; DOI : 10.1249/MSS.0b013e3181ed61a3) has shown that a window of 60 min sharply increases the accuracy of classification of wear time in comparison with a window of 20 min.

Lines 154: Maybe the authors could precise that MPA and VPA refer to “minutes” spent in MPA/VPA?

Lines 165-166: The authors cited the work by Craig al. (2003) to support the use of the German version of the IPAQ in the present study, but the German version was not used in the cited work. Do the authors know if the German version has been previously validated? Again, the authors could provide a more accurate view of what is known and unknown.

The Statistical analysis section could be improved. First, it was surprising to read information regarding analysis for the main study aim after those for the secondary aims. Second, why did the authors use only correlation analysis to analyze stability (and not, for example, the mean or the median change in the error between T0 and T1)? This could be interesting information regarding the magnitude of the differences between T0 and T1.

Lines 224-225: “PA measured by accelerometry at T0 shows a significant relationship with PA measured by accelerometry at T1 with the lower confidence limit of rs greater than .5 for adult’s MPA at both epoch lengths”. Are the authors dealing with adults only for this sentence?

Supplemental Tables 3/4: This is not “criterion validity” because there was not really a good criterion for all comparisons in the present study. The term “agreement” could be a more appropriate option to deal with all the comparisons that the authors performed.

For some results, there was both a “significant correlation” with p values <=0.05 and also have a 95% confidence interval that included 0 (eg, VPA: IPAQ vs Acc10 in adults, sup. Table 3). Can the authors comment this and make clearer what inferential statistics they mainly used to make conclusions about their work?

Figure 2 should be improved. Indeed, bar graphs are not very informative or worse, they can be misleading about the distribution of the data (Weissgerber et al., 2015; DOI: 10.1371/journal.pbio.1002128). Thus, simple bar graphs should be avoided. A much better version of this figure could be obtained using rainclouds plots (Allen et al., 2019: DOI doi.org/10.12688/wellcomeopenres.15191.1), but at least the figure should show the individual data. Moreover, it is surprising to have so much correlations in this paper with no scatter plot.

Section 4.1.1.: I have been surprised that the change in the actual physical behavior has not been evocated as a possible source of variability in the IPAQ scores. Can the authors comment on this?

Lines 395-396: “Finally, the use of validated measurement tools strengthened the explanatory power of results and allowed for comparison with existing research.” I suggest being more nuanced as the validity of the methods used in the present study (German version of the IPAQ, recent versions of the movisens devices) have not been related to appropriate literature in the paper, for now.

Line 399: “Spirometry”. Did the authors mean “indirect calorimetry”? Moreover, I am not sure that saying “using multiple accelerometers (e.g. hip, tight, and arm)” allow getting a “true value of PA” (line 397) is correct.

MINOR COMMENTS

Lines 44-45: It could be interesting to precise depending on what the correlations may be low to strong (correlations are likely to be different according to the studied outcome for example).

Lines 111-112: It is not clear for me what methods were considered for validity assessment. What was the gold standard?

Lines 130-13, it is written “T0 and T1 (which were at least four weeks apart).” and in the supplemental Table 1/2 captions,  it is written “two measurements three weeks apart (T0 and T1)”. What is the correct information?

Line 198: “r” = “rs”?

It could have been interesting to propose a figure or a table allowing the comparisons of the correlations used for validity at T0 and T1. For now, it is very difficult to make comparisons because the results are in different tables (cf. supplemental Tables).

The reference 14 is not complete.

Reviewer 2 Report

This paper presents a comparative study between the measured physical activity (PA) and the self-reported one in both adults and children. The study requested a group of 32 adults and 32 children to daily wear a 3-axial accelerometer during at least 8h in a two-week period together with self-reporting the physical activity on a dairy and answering the IPAC questionnaire. Results were then analyzed using stability, reliability, and validity criteria. Overall, the accelerometer measurements showed reliable, stable, and valid results while the self-reported ones yielded mixed results. 

Overall, the paper presents an interesting study showing that the actual PA might be quite different from the subjective perceived one. The paper is well written, it is easy to read, and to follow. Nevertheless, I recommend a major revision with following remarks:

A.    Content
a.1) The sensor content is sparse. Taking into account that the journal targets Sensors and their applications, I recommend expanding this section: include a photo of a user wearing the device, signal samples for light, moderate, and vigorous PA, and the main features of the MET estimation model for classifying PA intensity.

a.2) It is not clear what the SMARTFAMILY study is. Please clarify.

a.3) It is difficult to mentally visualize the content in tables 4 and 5. A graphic representation may be more suitable.

a.4) Further applications of the present study are recommended in the Conclusion (i.e., which fields could benefit/should be aware of the difference between the actual and perceived PA?)

B.    Format
b.1) Line 36: It is strange to cite a blog post. Any journal/conference paper that can replace this reference?
b.2) Line 225: .5 --> 0.5

Round 2

Reviewer 1 Report

Response 11: The correlation is indeed an important tool to compare two methods but an issue is that it cannot tell you the importance of the difference between the considered methods. For example, you can have a perfect correlation while having a systematic important difference between the two methods (such a situation could be approximatively reached if the random error were very low). The information delivered by a Spearman correlation is all the more poor that it assesses the relationship between ranks, not values, which occults the magnitude of the potential differences between the methods. To have some information on the difference between the methods, the authors could simply calculate the difference between the two considered methods at T0 and T1, and compute the mean or the median of the individual differences found between the errors related to T0 and T1, with the corresponding confidence interval. Basically, it could inform on how much the difference between the 2 considered methods changes between T0 and T1. For a given participant, this change could  be computed as follows:

Error at T1 minus Error at T0

The mean or median change would be the mean or the median of these individual differences. The mean/median change in the error is a suggestion but other methods can be used to inform on the change in the magnitude of the error found at two different time points.

Response 14: I did not understand this response: “Therefore, the confidence interval varies for each calculation”. Did the authors mean that the confidence interval found after bootstrapping changed after every analysis for a same analysis? If it was the case, the authors should fix the seed using the set.seed() function in R before each bootstrapping procedure to force R to provide the same results when performing the same analysis. Anyway, particularly if confidence intervals and p-values led to different conclusions, the authors should indicate in the statistical section what was the (a priori defined) statistics used to claim significance.  

Response 15: I appreciate the effort made by the authors to provide a much more informative figure. We can now see that some variables seemed approximatively normally distributed while other were very skewed due to outliers. This now allows the reader to be ensured that a non-parametric correlation was indeed an appropriate choice (at least, if no data transformation was performed). Unfortunately, in the current version of the revised manuscript, the quality of the figure was pretty poor. It would be preferable to increase the figure resolution for the final version.

Reviewer 2 Report

The paper has undoubtedly improved from its previous version. I appreciate that my remarks were taken into account. In particular, the sensor description and the possible application fields of this study complete the research work. 

I have no further major comments or remarks. However, before marking a full accept, I have some final remarks:

a) Line 45: Your new text mentions MPA and VPA without explaining what they stand for. MPA is defined in Line 114 while VPA in line 243.
b)  Please note that the quality of Figures 1 and 2 are very low. Labels in them are illegible.
